# Transforming Industrial Maintenance with Thermoelectric Energy Harvesting and NB-IoT: A Case Study in Oil Refinery Applications

**DOI:** 10.3390/s25030703

**Published:** 2025-01-24

**Authors:** Raúl Aragonés, Joan Oliver, Carles Ferrer

**Affiliations:** 1Department of Microelectronic and Electronic Systems, Universitat Autònoma de Barcelona, Bellaterra, 08193 Barcelona, Spain or raul.aragones@aeinnova.com (R.A.); joan.oliver@uab.cat (J.O.); 2R&D Department, AEInnova—Alternative Energy Innovations, S.L., AEInnova, Terrassa, 08224 Barcelona, Spain

**Keywords:** energy harvesting, thermoelectricity, LCA, carbon footprint, LoRaWAN, edge computing, NETZERO, energy-intensive industry

## Abstract

Heat-intensive industries (e.g., iron and steel, aluminum, cement) and explosive sectors (e.g., oil and gas, chemical, petrochemical) face challenges in achieving Industry 4.0 goals due to the widespread adoption of industrial Internet of Things (IIoT) technologies. Wireless solutions are favored in large facilities to reduce the costs and complexities of extensive wiring. However, conventional wireless devices powered by lithium batteries have limitations, including reduced lifespan in high-temperature environments and incompatibility with explosive atmospheres, leading to high maintenance costs. This paper presents a novel approach for energy-intensive and explosive industries, which represent over 40% of the gross production revenue (GPR) in several countries. The proposed solution uses residual heat to power ATEX-certified IIoT devices, eliminating the need for batteries and maintenance. These devices are designed for condition monitoring and predictive maintenance of rotating machinery, which is common in industrial settings. The study demonstrates the successful application of this technology, highlighting its potential to reduce costs and improve safety and efficiency in challenging industrial environments.

## 1. Introduction

Air pollution and climate change are among the most pressing global issues of our time. Jos Lelieveld et al. [1] highlight the severe health risks posed by air pollution, which is responsible for approximately 7 million deaths annually worldwide and reduces global life expectancy by an average of 2.2 years. Additionally, preliminary findings from the latest United Nations Climate Change Conference (COP29), held in November 2024, indicate from the IPCC and associated studies significant risks related to sea level rise and global temperature increases by 2100. Under high-emission scenarios, global temperatures could rise by approximately 3.5 °C to 4.4 °C by the end of the century. This would likely result in a global mean sea level rise of between 30 cm and 1 m, with further contributions expected from the melting of polar ice sheets. The most extreme scenarios, including a collapse of the Antarctic ice sheets, could see a rise of up to 1.6 m by 2100 and potentially up to 9–10 m by 2300 if emissions remain unchecked [2,3].

Further, a direct link between carbon emissions and Arctic ice melting has been established. Dirk Notz et al. [4] report that Arctic ice has decreased by half over the past 40 years. During the United Nations Climate Conference held in New York on 19 September, Professor Dirk Notz from the Max Planck Institute for Meteorology in Hamburg, Germany, revealed that every ton of CO_2_ emitted into the atmosphere results in the loss of 3 m^2^ of Arctic ice [5,6], measured in square meters per capita, as presented in the Guardian Report.

Energy-intensive industries are recognized as a major source of greenhouse gas emissions. Over recent years, these industries have taken steps to address their environmental impact, improving productivity, efficiency, and profitability as part of the Industry 4.0 transformation. Emerging IoT technologies offer significant opportunities to further enhance industrial efficiency and promote sustainability. However, sectors such the as oil and gas and petrochemical industries face unique challenges, particularly in hazardous (ATEX) environments, where deploying large-scale IoT solutions remains complex [7]. According to market projections from Statista and IoT Analytics, the IoT industry has shown remarkable growth, with over 15 billion connected devices recorded in 2023. Forecasts predict that this figure will exceed 30 billion devices by 2030, driven by advancements in smart technologies and increased adoption across industrial sectors [8,9].

Industrial processes are responsible for the emission of large quantities of greenhouse gases, particularly carbon dioxide (CO_2_), methane (CH_4_), and nitrous oxide (N_2_O), as they require substantial energy for raw material transformation. 

A significant portion of industrial energy is lost as waste heat, and in the European Union, approximately 20–50% of the energy used in industrial processes is wasted in the form of heat, such as hot exhaust gases and cooling water [10]. Moreover, about 21% of the EU’s total annual energy consumption comprises energy lost as industrial waste heat, making this a critical area for energy efficiency improvements [11]. This highlights the substantial potential for waste heat recovery (WHR) technologies to reduce overall energy consumption and CO_2_ emissions across industries [12].

To address these challenges, this paper introduces an innovative Industrial IoT (IIoT) sensing technology designed to operate in industrial and hazardous environments. The proposed system is maintenance-free and battery-less, ensuring long-term reliability and sustainability. It incorporates wireless long-range communication, edge-computing capabilities, and ATEX certification, making it ideal for deployment in explosive atmospheres. The system also delivers low-latency data transmission. Initial experimental results indicate that this solution reduces environmental impact by up to 94% compared to conventional alternatives, while simultaneously cutting sensing costs by approximately 70%. These performance metrics are based on internal testing and benchmarking studies.

The remainder of this paper is structured as follows. Section 2 discusses the environmental and economic drivers that necessitate adopting sustainable sensing technologies in the oil and gas sector. Section 3 explores the potential of thermoelectric technologies for waste heat harvesting and conversion into electrical energy. Section 4 provides a detailed overview of the thermoelectric generator powering the edge-computing IIoT device. Experimental results, including the performance of the IIoT vibration sensor using the NB-IoT wireless protocol, are presented in Section 5, along with the system’s deployment in a Spanish oil refinery from Repsol. Finally, Section 6 summarizes the key findings and conclusions of this work.

## 2. Environmental Impact

Energy-intensive industries (EIIs) face a significant challenge in improving their operational efficiency, maintenance processes, energy consumption, and overall sustainability. While many IIoT devices rely on direct power connections, the increasing adoption of long-range wireless protocols such as LoRaWAN and NB-IoT—the latter being the focus of this study—offers a transformative solution for remote monitoring in areas where local power wiring is unavailable or impractical. These protocols not only enable reliable data transmission but also open the door to battery-free, heat-powered devices, contributing to enhanced energy efficiency and sustainability. 

This work focuses on NB-IoT, a protocol that enables direct cloud connectivity, facilitating advanced data analytics, including machine learning (ML) and deep learning (DL), often limited in traditional SCADA systems.

### 2.1. Space-Driven Technology to Power the Industrial Internet of Things

The Industrial Internet of Things (IIoT) has emerged as a critical enabler for achieving these challenges (operational efficiency, maintenance processes, energy consumption, and sustainability). It is widely regarded as one of the five key technologies poised to significantly transform industries in the coming years. Alongside technologies such as artificial intelligence (AI), blockchain, and robotics, IIoT is expected to play a critical role in driving the next wave of digital transformation. By enabling real-time data collection, improving operational efficiency, and enhancing customer experiences, IIoT is becoming increasingly integral to sectors like manufacturing, healthcare, and logistics.

Each IIoT device requires power in the range of 10–100 mW, with batteries currently serving as the primary energy source. However, the reliance on batteries remains a significant obstacle, particularly for applications that require miniaturized nodes or embedded energy. Miniaturization presents an effective, cost-efficient method for expanding the network of connected devices, adhering to the “smart dust” concept. This approach, however, imposes space constraints that necessitate smaller batteries with even more limited energy capacity. As a result, devices powered by primary batteries often have an operational lifetime limited to a few months, necessitating frequent battery replacements. This introduces substantial economic costs, environmental concerns, and logistical challenges. Accordingly, alternative compact, low-cost, and lightweight energy sources are required. One promising solution is energy harvesting from industrial waste heat using thermoelectric converters. The first application of this technology dates back to NASA’s deep space missions, such as the Voyager spacecraft launched in 1977 [13]. For these missions, converting into electricity using thermoelectric components (Peltier cells) provided a reliable power source, leveraging the available thermal energy from the environment. Since then, over 63 deep space missions have utilized this principle to power electronic systems in space.

### 2.2. Waste Heat Recovery Potential

To mitigate the significant energy loss in industries due to waste heat, we propose the implementation of a thermoelectric-powered system designed to efficiently recover and utilize this lost energy. This system is created by the Spanish company AEInnova (Alternative Energy Innovations). It is a spinoff of the Microelectronics and Electronics System department of the Autonomous University of Barcelona, recently recognized in November 2024 by the European Commission Innovation Radar as representing the most sustainable technology in Europe [14]. All authors are also founders of this company.

Its product, InduEye, offers a sustainable alternative to conventional battery-powered devices by addressing several key challenges:Eliminates the need for frequent battery replacements.Removes restrictions on edge-computing energy consumption.Facilitates the transmission of high-frequency data using long-range communication protocols.

In essence, waste heat (WH) emissions pose substantial economic and environmental challenges, as a large portion of energy used in industrial processes is dissipated as heat. Figure 1 illustrates the potential heat loss across various industrial sectors, highlighting the opportunity for sensor technologies powered by thermal energy capture to contribute to energy recovery and enhance operational efficiency.

The potential for energy recovery from waste heat is highly dependent on several interrelated factors, including the physical origin of the waste heat, which determines its characteristics, and the mode of heat transfer, whether through conduction, convection, or radiation. Additionally, the theoretical energy extraction potential is governed by the principles of the Carnot cycle, which defines the maximum conversion efficiency between heat reservoirs. The efficiency of heat extraction systems must also be considered, particularly the losses such as through self-heating during energy recovery processes. Furthermore, the technological and economic feasibility of designing and implementing waste heat recovery systems plays a crucial role.

To analyze energy sustainability in this context, two critical concepts are commonly used in the scientific field: exergy, which refers to the usable portion of energy that can be converted into work, and anergy, which denotes the portion of energy that cannot be converted into useful work.

The exergy content of waste heat can be determined using Carnot’s formula, which calculates the theoretical maximum efficiency of a heat engine based on two heat reservoirs. By applying the Carnot factor to waste heat temperatures, it is possible to estimate the corresponding technical work potential.(1)$$n_{max}=n_{c}=1-\frac{T_{low}}{T_{high}}$$

To define an appropriate temperature range for waste heat classification, three main categories are established: low temperature (LT), medium temperature (MT), and high temperature (HT). Low temperature refers to heat below 100 °C, medium temperature corresponds to the range between 100 °C and 300 °C, and high temperature applies to heat exceeding 300 °C.

As illustrated in Figure 2, based on current technologies and Carnot’s theoretical limitations, the maximum recovery potential is achieved at higher temperatures. This is because, at elevated temperatures, the efficiency of recovery systems improves, and the limitations imposed by Carnot’s performance become less restrictive, allowing for greater technological application and energy extraction.

By analyzing the temperature ranges by sectors, the actual available technology, and Carnot’s limitations, it is possible to know the limits of waste heat recovery technology. In this way, Figure 3 shows the waste heat recovery potential by sector [9].

Based on the temperature of waste heat, heat losses can be categorized into three ranges: high (above 400 °C), medium (between 400 °C and 100 °C), and low (below 100 °C). Each temperature range requires suitable waste heat recovery systems to maximize energy recovery.

An analysis conducted in [16] examines the distribution of residual heat across various industrial sectors, considering the source of the heat. In this context, wasted energy is evaluated using Carnot efficiency, where temperature is a key factor in determining the potential to recover waste heat as useful work. Table 1 classifies the different sources of waste heat according to the temperature ranges most compatible with thermoelectric harvesting technologies.

The current state of thermoelectric technologies enables operation at temperatures exceeding 600 °C. For industrial processes that reach temperatures above 1000 °C, the development of specialized heat exchangers is required. Ongoing R&D efforts in thermoelectric materials are also enhancing heat-to-power conversion efficiency, particularly at lower temperature ranges.

Waste heat conversion into electricity represents an attractive solution for industries due to its versatility, high value, and potential to reduce energy costs while improving overall efficiency. Table 1 identifies waste heat sources, such as gas exhausts, drying, and cooling processes, many of which fall within the 150–300 °C range—ideal for thermoelectric recovery technologies. AEInnova has conducted field tests for heat recovery in scenarios such as cement manufacturing, wineries, alcohol production, and steel manufacturing, confirming significant potential for industrial applications.

However, challenges persist, primarily due to the moderate efficiency of thermoelectric generators and long payback periods. Market studies [17,18] reveal that industries typically seek a return on investment (ROI) of under five years. Although rising energy prices may accelerate adoption, reducing system costs through scaling production and implementing more powerful thermoelectric cells will be essential for widespread market integration.

### 2.3. Environmental Effects of Lithium Batteries

The energy-intensive industries (EII) face significant challenges in improving operational efficiency, energy consumption, and sustainability. Among these, reliance on lithium batteries as the primary energy source raises environmental and safety concerns:**Raw Material Processing**: Battery production involves the extraction of resources like lithium, primarily sourced from the lithium triangle (Chile, Bolivia, Argentina). This process causes environmental issues such as excessive water consumption, ecosystem disruption, and waste generation.**Production and Charging**: Over 85% of global battery production occurs in China, where coal-based energy (82%) [19] dominates, increasing the carbon footprint of battery manufacturing.**Waste Management**: The improper disposal of lithium batteries, with around 25 million discarded annually, poses fire hazards and releases toxic gases harmful to people 1and the environment.

To enable the sustainable adoption of IoT technologies in EII, it is essential to address these challenges through responsible material sourcing, improved lifecycle management, and mitigation of environmental impacts.

### 2.4. Lithium Batteries in Explosive Environments

Industries such as oil and gas, chemicals, steel, cement, glass, plastics, and mining encounter significant challenges when implementing common battery-powered IoT devices, mainly due to the limitations of lithium-ion batteries in hazardous environments. These batteries experience considerable degradation in high-temperature settings. For instance, an IoT device powered by a lithium battery at temperatures above 85 °C is estimated to lose over 60% of its lifespan compared to a device running at 20° [20]. Moreover, these batteries are unsuitable for use in explosive environments (ATEX) due to safety concerns. They also depend on low-range communication protocols like WirelessHART and ISA100, which require substantial investments in wireless infrastructure, including repeaters and gateways.

To address these challenges, this paper offers a cost-effective Industrial IoT solution, initially designed to monitor rotating machinery (electric machines), effectively reducing monitoring costs in such industries.

## 3. Technological Approach

The authors propose a cost-effective and environmentally friendly solution focused on energy harvesting technology to power advanced sensing solutions for predictive maintenance in industrial processes, a solution called InduEye (Industrial Eye). 

### 3.1. InduEye Description

InduEye is a battery-less and heat-powered IoT device designed to monitor various industrial processes, such as vibration in rotating machines or steam leaks in steam traps. It operates using a thermoelectric generator to harvest energy from waste heat, a monitoring system, and a wireless chipset (LoRa or NB-IoT) to transmit data to the cloud. This system eliminates the need for cables or batteries, making it suitable for a wide range of industries.

InduEye consists of three components (Figure 4):A thermoelectric generator and energy harvester.An edge-computing device for data acquisition, processing, and transmission.A sensory system, capable of measuring vibration, temperature, and sound.

Depending on the installation, these components can be integrated into one device or used separately, offering flexibility for different operational needs.

### 3.2. Thermoelectric Generator

The thermoelectric generator is the device that converts heat into electricity through the Peltier effect. This is shown in more detail in Figure 5.

The HEAT-RU (heat recovery unit) is composed of the following elements:Top (in blue): an aluminum alloy A6060 hot side radiator, with a heatsink to create the corresponding delta T between the sides of the module. It is cooled by convection air.Middle top (in green): isolating material, to guarantee heat isolation between both sides.Middle lower: thermoelectric generators (TEGs) from Kryotherm TGM199.Bottom: housing and heat transfer. This provides the adequate device structure and transfers the heat of the hot surface to the TEGs.

#### 3.2.1. Thermoelectric Model

In [21], the model, simulation, and comparison with real data for the HEAT-RU are presented. According to the authors, accurately modeling these systems poses significant challenges. The HEAT-RU consists of passive components, including the collector, heatsink, and other adjusting elements, as well as the thermo-generator module (TGM), which serves as the active component that converts harvested heat into electrical energy.

The HEAT-RU model interconnects each passive component, described through their thermal resistance and heat capacitance, with the TGM, whose behavior is influenced by the temperature differential between its two sides. The complexity of modeling each component depends largely on its geometry and construction. Heatsinks, for example, are particularly difficult to model due to the intricate geometry of their base and fins, where the efficiency is influenced by the temperature gradient between the fins and the surrounding environment. Additionally, thermal contact resistance between components, which is influenced by surface roughness, further complicates the modeling process.

The TGM adds another layer of complexity, as manufacturers often do not provide detailed thermal and electrical properties of the thermoelements. To address this issue, the “effective parameters” approach introduced in [22,23] is applied, allowing the TGM to be modeled in terms of resistive paths, thermal capacitances, and heat sources or collectors. Figure 6 illustrates the HEAT-RU model, which represents the heat transmission channel within the system.

#### 3.2.2. Peltier Cell Model Using Effective Material Properties

The overall performance of the HEAT-RU thermoelectric converter is determined by the efficiency of the thermoelectric generator (TEG) cell in converting thermal energy into electrical energy. The operation of TEG cells is governed by several thermoelectric effects, including the Seebeck, Peltier, Thomson, and Joule effects.

The Seebeck effect is primarily responsible for electricity generation within the cell. This phenomenon occurs when two different semiconductor materials, **A** and **B**, are joined at their ends by a conductive material and subjected to a temperature difference between the joints. This temperature gradient causes a flow of charge carriers, resulting in the conversion of heat energy into electrical energy. The electromotive force generated by the Seebeck effect is quantified by the Seebeck coefficient, as expressed in Equation (2).


(2)
$$\frac{\partial E_{T}}{\partial T}=\alpha_{a}-\alpha_{b}$$


The Peltier effect refers to the heating or cooling that occurs when an electric current flows through the junction of two different materials, operating inversely to the Seebeck effect. This effect causes heat to be absorbed or released at the junction, depending on the direction of the current flow. The rate of heat transfer associated with the Peltier effect, denoted as ${\dot{Q}}_{Peltier}$, is defined by Equation (3).


(3)
$${\dot{Q}}_{Peltier}=\pm IT\left(\alpha_{a}-\alpha_{b}\right)$$


The Joule effect describes the heating of a material when an electric current flows through a conductor, resulting in the dissipation of electrical energy as heat. The heat flux produced by this effect, denoted as ${\dot{Q}}_{Joule}$, is expressed in Equation (4), where *R*_0_ represents the internal electrical resistance for this phenomenon.


(4)
$${\dot{Q}}_{Joule}=R_{0}I$$


The Thomson effect occurs when an electric current flows through a single material that experiences a temperature gradient along its length. Depending on the Thomson coefficient, this effect can result in either heat absorption or generation. The resulting heat flux is defined by Equation (5) and is influenced by the Thomson parameter σ.


(5)
$${\dot{Q}}_{Thomson}=-\sigma I\Delta T$$


The conversion efficiency of a Peltier cell is defined by the parameter Z (units of 1/K) or the dimensionless figure of merit Z_T_, where T represents the average temperature of the cell. The Z factor characterizes the relationship between the energy supplied to the load and the heat transfer occurring within the cell. The maximum efficiency, expressed in terms of Z_T_, is given in Equation (6a) and is widely used to evaluate thermoelectric devices.

The value of Z_T_ depends on the average temperature T and the thermoelectric material parameters: the Seebeck coefficient (α), electrical conductivity (σ), and thermal conductivity (κ). Additionally, the term η_max_ = 1−T_H_/T_C_ (Equation (6b)) corresponds to the Carnot efficiency. In [21], these equations are applied to simulate and model the performance of the HEAT-RU system.(6a)$$ZT=\frac{\sigma \alpha^{2}T}{\kappa }$$(6b)ηmax=TH−TCTH 1−ZT¯−11+ZT¯−TCTH

To model the behavior of the thermoelements in the thermoelectric generator module (TGM), it is fundamental to determine parameters α, σ, and κ of the Equations (2)–(6). Nevertheless, these parameters are not usually given by the manufacturer. In [22], these thermocouple parameters are calculated (and named *Effective Parameters* α*, κ*, *ρ**, and Z*) using the maximum characteristics of power W_max_, current I_max_, electromotive force V_max_, efficiency η_max_, and number of thermoelements n of the cell, given in the TGM datasheet. 

#### 3.2.3. DC/DC Power Converter 

The power electronic circuit converts the energy harvested by each thermoelectric generator module into a stable 3.3 V DC output, which powers the IIoT device. As shown in Figure 7, the power electronics system optimizes the TGMs’ performance, ensuring that they operate at maximum efficiency to harvest the most thermal energy possible. The energy is stored in a 5 F supercapacitor, capable of providing sufficient power for up to five complete acquisition and communication processes.

## 4. Edge Node with NB-IOT

In this section, we will present the technological approach for the most advanced wireless IIoT device on the market, which is not powered by a battery and incorporates real-time processing with edge-computing capabilities.

### 4.1. The Battery-Less IIoT Vibration Monitor

Typically, IIoT wireless sensor devices require power in the range of 30–150 mW. Batteries are currently the main energy source for powering remote and portable IoT nodes. However, battery storage remains an unresolved issue and a major challenge that hinders the development of certain applications, especially when miniaturized nodes and real-time data processing are required, as demanded in several industries.

The space limitations of small-sized nodes require smaller batteries, which have an even more limited energy capacity. As a result, in many applications, primary batteries can only provide energy for a limited period corresponding to the device’s lifetime (from a few months to 1–2 years, depending on the data frequency required). This necessitates regular battery replacements, which incur significant economic costs, environmental impacts, and logistical challenges, particularly in large industrial settings.

### 4.2. Technology Challenges for Wireless IIoT

Additionally, industrial monitoring requires that all processes be controlled within the same plant, with full control of the systems and no interaction with third parties.

In summary, the drawbacks that define the battery lifetime (in months) in wireless IoT devices concern the following:Battery energy storage (mAh).Environmental conditions (temperature and humidity).Power consumption of electronic components:
○DC/DC converter efficiency.○CPU processor consumption:
▪Operational frequency (MHz).▪Power-saving modes (sleep, ultra-sleep, slow-down, standby, etc.)▪Edge-computing algorithms.▪Firmware optimization.
○Sensor power consumption.○Sensing conditioning electronic components (sample and hold, amplifiers, filters, etc.).○Wireless communication protocols.


These limitations are highly restrictive when manufacturing battery-powered IoT devices, especially in hazardous environments (EX/ATEX) where lithium cannot be used. 

In this energy-harvesting-powered edge-computing device, the continuous power supply provided by the thermoelectric generator eliminates the common restrictions of wireless, battery-powered devices. As a result, it can power and read data from high-energy-consumption sensors, such as in the range of 4–20 mA, while also supplying power to them. It enables long-range wireless communications without energy limitations and supports high-CPU-frequency-demanding algorithms for sensor data processing, such as FFT, digital filters, and machine learning, enabling edge computing. Additionally, it delivers data almost on-demand every few seconds with a high-speed throughput of up to 1 Mb/s. 

### 4.3. Long-Range Wireless Protocols Comparison

One of the most important factors to consider in wireless sensor networks is the selection of the wireless protocol. This choice primarily determines node energy consumption, the maximum range and the number of nodes that can connect to the same gateway or repeater, as well as data rate and latency. It also impacts network costs, including whether bands are licensed or unlicensed, infrastructure expenses, and the encryption of node data for security.

For very large facilities and real-time critical processes, the NB-IoT wireless protocol stands out as the ideal choice due to its unique balance between range, data rate, and moderate power consumption Table 2. Compared to Zigbee, Wi-Fi, Bluetooth, LoRaWAN, Sigfox, and LTE-M, NB-IoT offers superior scalability and extended coverage, reaching up to 22 km, which makes it particularly suitable for expansive sites such as oil refineries. Unlike Zigbee, Wi-Fi, and Bluetooth, which are primarily designed for short-range applications and operate in unlicensed frequency bands, NB-IoT leverages licensed LTE bands, ensuring better signal integrity and interference resistance in industrial environments. Additionally, its data rate (256 kbps) is higher than that of LoRaWAN and Sigfox, facilitating the transmission of richer datasets while maintaining lower latency, a critical factor in real-time monitoring. While LTE-M provides a higher data rate, its shorter range and higher power requirements make it less suitable for extremely large facilities where power efficiency is a priority. As a result, NB-IoT’s combination of long range, reliable performance, and moderate power usage makes it a top choice for large-scale industrial deployments [24].

### 4.4. Edge-Computing Node Internal Architecture

Figure 8 and Figure 9 show the NB-IoT sensing node consisting of the following components:Communications hardware: includes a 5dBi antenna, the Quectel BG96 NB-IoT UART module, and a SIM.Power electronics: features a DC/DC converter to power the external sensor, an alternative energy buffer, energy management circuitry, and the SPI bus interface for the internal three-axis vibration IMU from STMicroelectronics.Programmable system on chip: equipped with an Infineon PSoC 4 32-bit processor, Flash and RAM memory, analog and digital FPGA, and communication modules such as SPI, I2C, and CAN bus.

### 4.5. Acquisition and Processing and Communication Flow

To determine the data latency as well as the dimensions of the energy buffer, it is necessary to first know all the energy that can be harvested to attend to the system energy demand.

Data latency can be calculated according to the power generation from the thermoelectric unit and the power for the edge-computing algorithm considering up to 1 data package per minute (at 150 °C).

To perform all the processes, a flow diagram of five time slots or states has been considered. Each time slot has its associated duration and power consumption that will determine the energy factor cost required to perform it (Figure 10), as plotted in Section 5.2. 

### 4.6. Cloud Computing Architecture

This subsection describes the complete wireless and cloud computing infrastructure used in the system, which leverages both Vodafone’s carrier network and Amazon Web Services (AWS) for cloud computing services. As shown in Figure 11:

The NB-IoT communication system does not require a local gateway, as it is designed for direct communication between the end nodes and the server. The data transmitted are minimal and currently encrypted using AES 128, considering the security provided by the Vodafone network.

The transport protocol used in NB-IoT is UDP, meaning a UDP server is necessary to receive the data. A program has been deployed on our server to receive the UDP packets sent by the NB-IoT nodes, group them, and then publish them to an MQTT server hosted on AWS (IoT Core). Communication with IoT Core is secured via TLS, utilizing AWS self-signed certificates. On the receiving side, a program (AEInnova Packet Forwarder) subscribes to this MQTT server to collect the data and insert them into the DAEVIS (Dynamic AEInnova VISualizer) IoT web dashboard.

Note: The architecture shown in Figure 11 targets a cloud-first approach using the NB-IoT protocol, which may face integration challenges with existing SCADA systems based on ISA95. The current design focuses on remote data acquisition for cloud-based analytics but can be adapted for local SCADA integration using additional industry-standard protocols (e.g., OPC UA, Modbus TCP, EtherCAT) in future iterations.

## 5. System Deployment 

In this section, we present the main benchmarks and results from the characterization of the complete solution (thermoelectric generator, IIoT device, and sensor).

### 5.1. Thermoelectric Generator Characterization

Figure 12 presents data recorded by the thermoelectric generator. It displays the harvested power, converted to 3.3 VDC by the DC/DC converter module. The ambient temperature in the facility was approximately 35 °C. 

Table 3 summarizes the data extracted from Figure 11, correlating it with the pipeline’s temperature where heat is being harvested. *Thot* represents the temperature reaching the hot side of the Peltier cell, while *Tcold* corresponds to the cold side. Notably, the HEAT-RU design achieves a temperature difference (Δ*T*) ranging from 43 °C to 57 °C. Lastly, the table’s final row displays the power generated by the module, which ranges from 780 mW to 1.05 W of DC power.

### 5.2. Edge Node Power Characterization

In this subsection, the IIoT node is characterized from the power-consumption point of view. For this characterization, two different communication methods have been compared (TCP and UDP). In particular, the data bridge solution uses TCP and UDP protocols that have different impacts in power consumption. While UDP gives the lowest power consumption, TCP presents a higher robustness. 

In particular, the IoT Core module from Amazon Web Services uses MQTT + TLS or HTTPS protocols which require x.509 authentication certification. TLS (Transport Layer Security) is a cryptographic protocol designed to provide communications security over any network [25]. The NB-IoT protocol can handle communications with and without the security provided by the carrier. 

When security is delegated to the carrier, data can be transmitted using UDP or MQTT protocols. If security is needed and it is not delegated to the carrier, then the right option is MQTT + TLS. 

For this pilot, three experiments have been performed to evaluate energy impact, as shown in Figure 13. 

Results comparing NB-IoT and LTE-CATM1 (both protocols are implemented inside the Quectel BG96 chipset) are represented in Table 4, and proportions for UDP and MQTT + TLS in Figure 14.

Compared with a previous work [21] using LoRaWAN, with an equivalent monitoring application, the energy profile was significantly lower due to the lower chipset consumption (Table 4).

The previous plots clearly demonstrate that the NB-IoT protocol offers significant energy savings compared to LTE-CATM1. Similarly, UDP provides considerable energy savings compared to MQTT-TLS. Figure 15 illustrates the normalization of both protocols for comparison.

### 5.3. Pilot Installation in an Air Compressor

The complete system has been installed to monitor vibrations in an air compressor at a Spanish oil refinery operated by REPSOL SA (Figure 16). The NB-IoT protocol, currently adopted by Spanish carriers Telefónica and Vodafone, enables data transmission. Vibration data are processed by converting measurements from the acceleration domain (in g) to the velocity domain (RMS in mm/s), in compliance with the ISO 10816 standard for predictive maintenance [27]. The analysis covers a frequency range from 1 Hz to 1000 Hz across three axes. Figure 16 illustrates the placement of the system components in the installation. In the figure, the thermoelectric generator (installed in an oil cooling pipeline) is shown in the left-hand box, the NB-IoT device at the bottom, and the three-axis vibration sensor in the right-hand box.

### 5.4. Data Representation

Figure 17 shows the registration of the compressor’s vibrations for a period of four days in Repsol’s Puertollano Oil Refinery (Spain), only operating 8 h per day, from the 14 to 17 October 2024. Data are presented on the AEInnova cloud visualizer tool called DAEVIS (Dynamic AEInnova VISualizer). The figure displays warning vibrations in the compressor for all four days (overpassing the threshold of 1.2 mm/s).

The monitored machine is classified as *Machine Group 4*, with a rigid base on the ground. From the data obtained, the health of the machine can be known according to the ISO Standard, which defines the critical speeds as:Healthy machine: from 0 mm/s to 1.4 mm/s.Short-term operation allowable: from 1.4 mm/s to 2.3 mm/s.Vibrations cause machine damage: from 2.3 mm/s to unlimited.

### 5.5. Related Work

To provide a comprehensive market analysis and assess the competitive landscape, Table 5 compares existing vibration monitoring technologies with our proposed solution. This comparison includes leading manufacturers offering similar systems and evaluates key parameters such as technological approach, functional features, target industries, and communication protocols. Notably, our solution is the only vibration sensor on the market that utilizes the NB-IoT standard for communication. Competing devices primarily rely on battery-powered systems integrated with wireless communication protocols or on wired solutions (e.g., Modbus). However, due to the energy demands of the NB-IoT protocol, battery-powered devices cannot guarantee data transmission beyond one week of operation without frequent battery replacements or recharges.

Our technology differentiates itself through its innovative energy-harvesting system, which uses residual heat as a power source instead of traditional batteries. This approach not only eliminates the limitations of finite energy sources but also aligns with sustainability goals by reducing waste and environmental impact. Furthermore, the integration of NB-IoT for long-range wireless communication provides a significant advantage over traditional short-range systems like WirelessHART or proprietary wireless protocols, which require substantial investment in gateways and infrastructure.

By combining energy harvesting with long-range connectivity, our solution sets a new standard for industrial monitoring, offering a sustainable, cost-effective, and scalable alternative to conventional technologies.

## 6. Conclusions

This study presents an innovative solution that uses waste heat to power IIoT devices in challenging industrial environments, such as oil refineries and chemical plants. The proposed system uses thermoelectric technology to eliminate the need for batteries, significantly reducing maintenance costs while ensuring compatibility with explosive atmospheres (ATEX). A key benefit of the NB-IoT protocol is its ability to provide long-range, low-power communication, which is particularly advantageous in large facilities like refineries where extensive wiring is impractical. The protocol ensures efficient data transmission with low latency, scalability, and high reliability, enabling effective real-time monitoring across large-scale industrial operations.

Furthermore, an in-depth study of the thermoelectric generator’s energy production was conducted, showing that the power generated is well-suited to meet the energy demands of the IIoT node. This ensures consistent and sustainable operation of the system without relying on external power sources. Additionally, the study compared different data transmission methods, including MQTT + TLS and UDP, assessing their impact on power consumption. It was found that UDP offers significant energy savings over MQTT + TLS, making it a more efficient option for low-power, long-range communication in IIoT applications.

While the primary focus of this study is on remote data transmission for cloud-based analytics using NB-IoT, we acknowledge that most critical industrial environments rely on direct SCADA integration due to security concerns under the ISA95 hierarchy. Future developments could explore multi-protocol compatibility to facilitate both cloud and on-premises data management.

The results demonstrate the potential of this solution to revolutionize energy-intensive industries by enhancing energy efficiency, reducing environmental impact, and improving safety and operational efficiency. The real-world deployment of this technology, particularly in a Spanish oil refinery, confirms its capacity to address critical industrial challenges, offering a sustainable and cost-effective approach to predictive maintenance and condition monitoring.

## Figures and Tables

**Figure 1 sensors-25-00703-f001:**
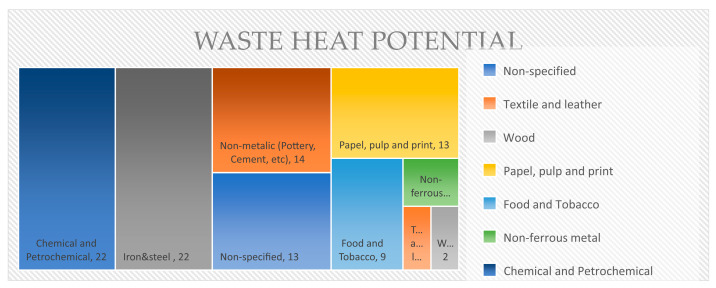
Waste heat potential per industrial sector (EU) [15].

**Figure 2 sensors-25-00703-f002:**
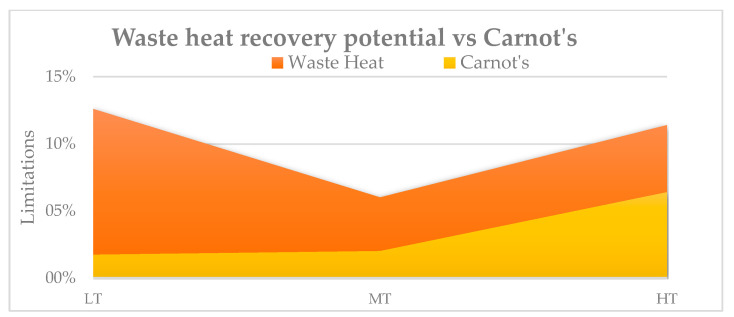
WH recovery potential (low temperature to high temperature vs. Carnot’s limitations.

**Figure 3 sensors-25-00703-f003:**
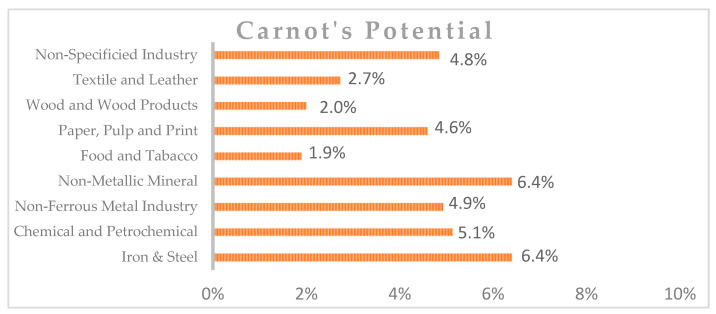
Waste heat recovery potential by sector.

**Figure 4 sensors-25-00703-f004:**
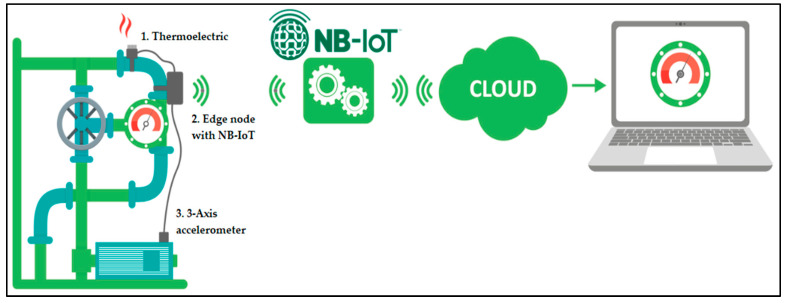
The NB-IoT scenario.

**Figure 5 sensors-25-00703-f005:**
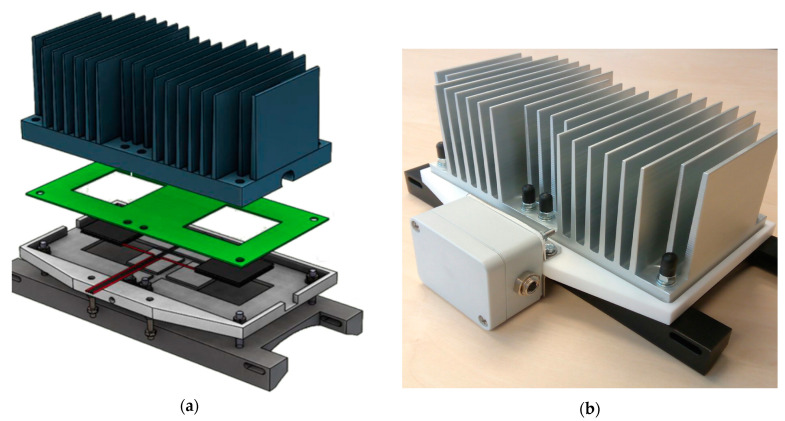
HEAT-RU: the thermoelectric module. (**a**) Equipment breakdown (**b**) final device.

**Figure 6 sensors-25-00703-f006:**
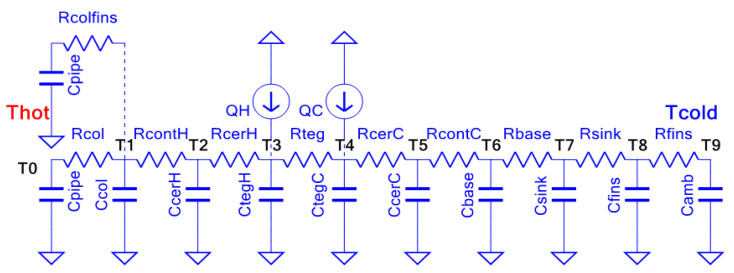
Model of HEAT-RU thermoelectric generator.

**Figure 7 sensors-25-00703-f007:**
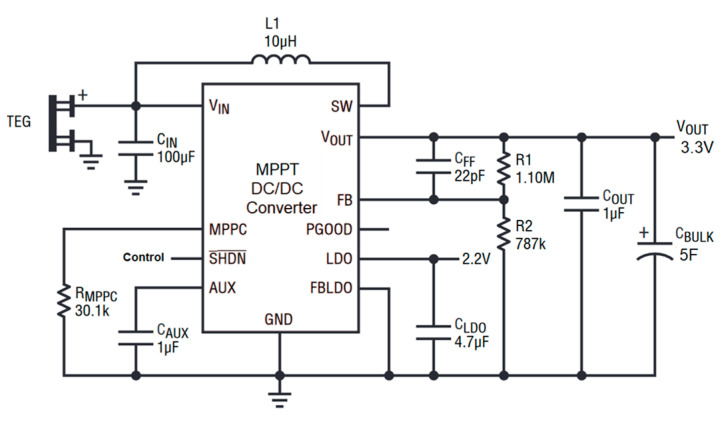
DC/DC power converter for the thermoelectric power generator.

**Figure 8 sensors-25-00703-f008:**
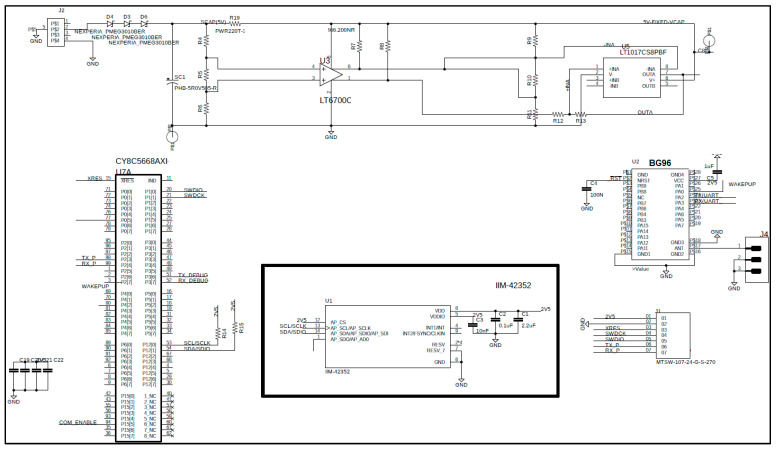
Schematics. Processing and communicating. Sensing board in the middle.

**Figure 9 sensors-25-00703-f009:**
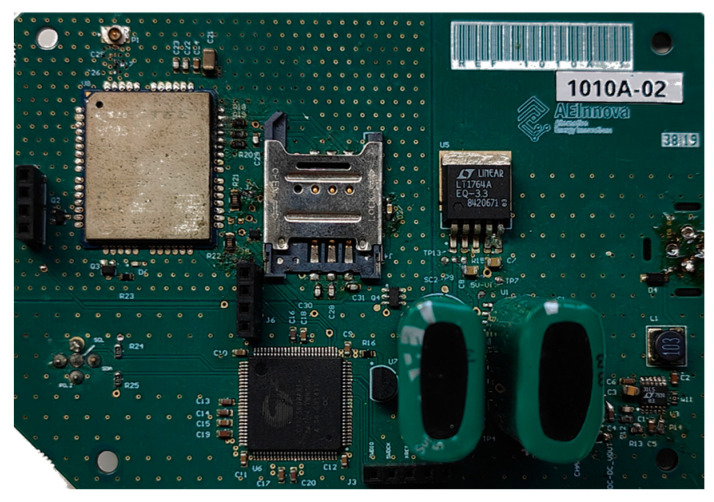
Full PCB with DC/DC power converter and processing and communicating module.

**Figure 10 sensors-25-00703-f010:**
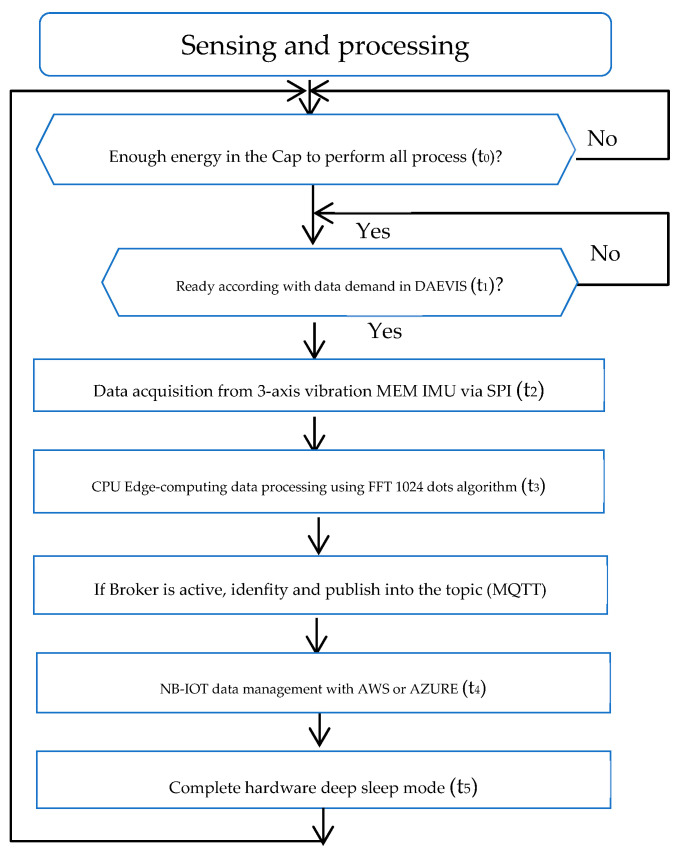
Flow diagram of the sensing/processing and communicating process.

**Figure 11 sensors-25-00703-f011:**
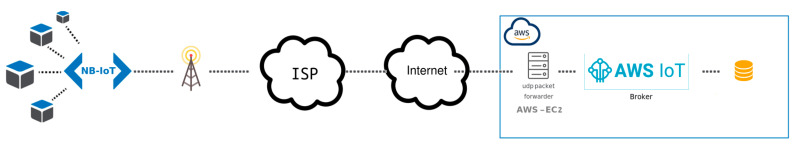
Complete wireless and cloud computing architecture.

**Figure 12 sensors-25-00703-f012:**
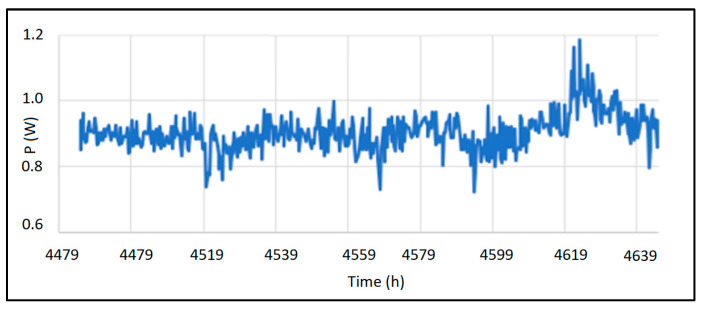
Complete wireless and cloud computing architecture.

**Figure 13 sensors-25-00703-f013:**
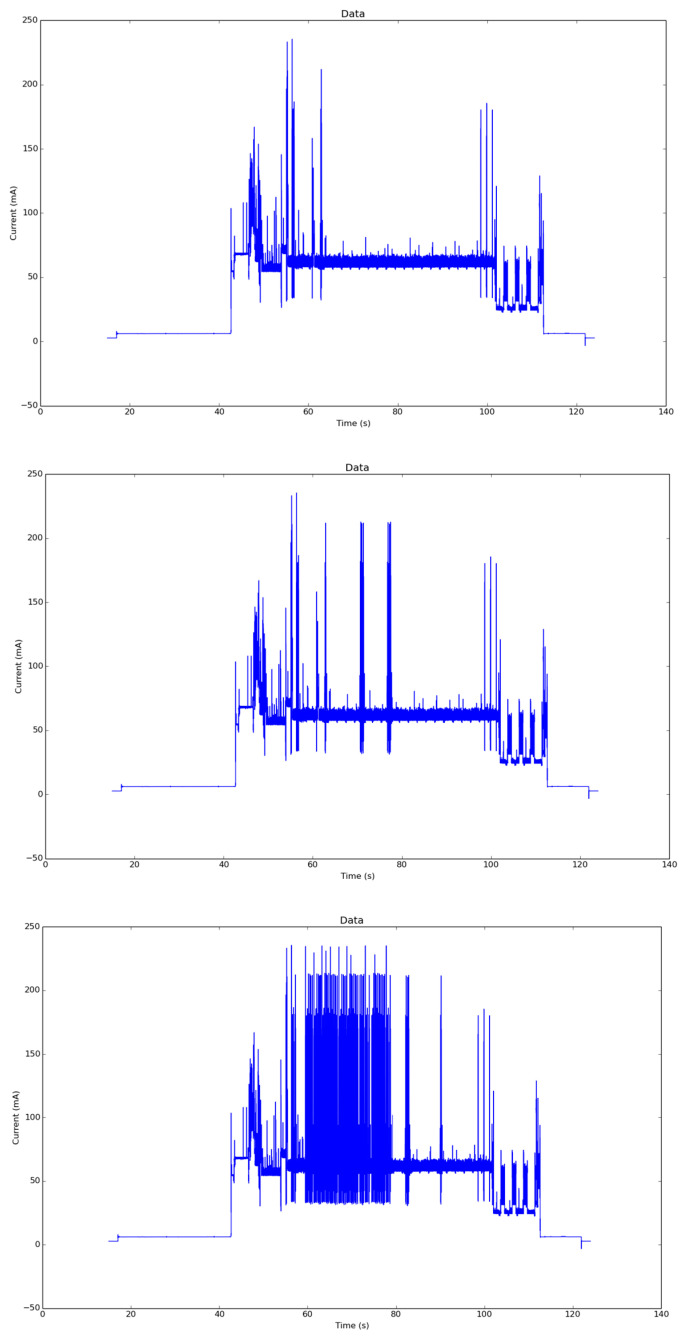
Top: UPD energy profile. Center: MQTT energy profile. Bottom: MQTT TLS energy profile.

**Figure 14 sensors-25-00703-f014:**
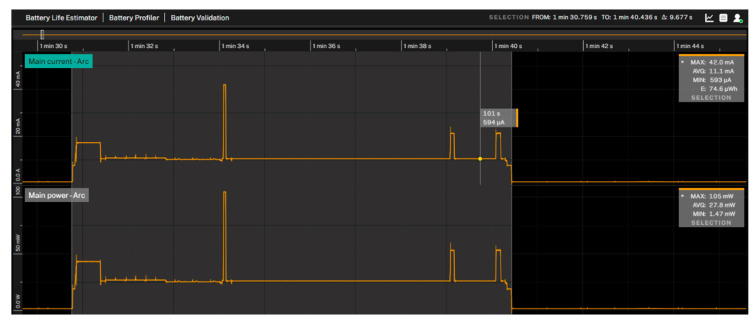
LoRaWAN energy profile [26].

**Figure 15 sensors-25-00703-f015:**
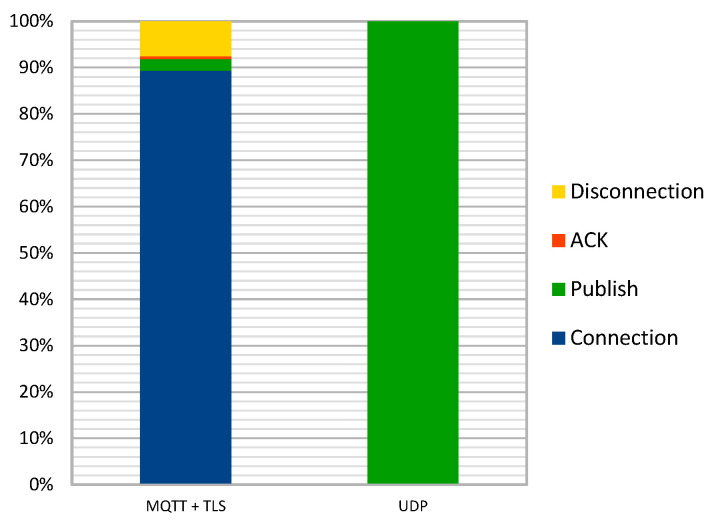
MQTT + TLS normalized versus UDP.

**Figure 16 sensors-25-00703-f016:**
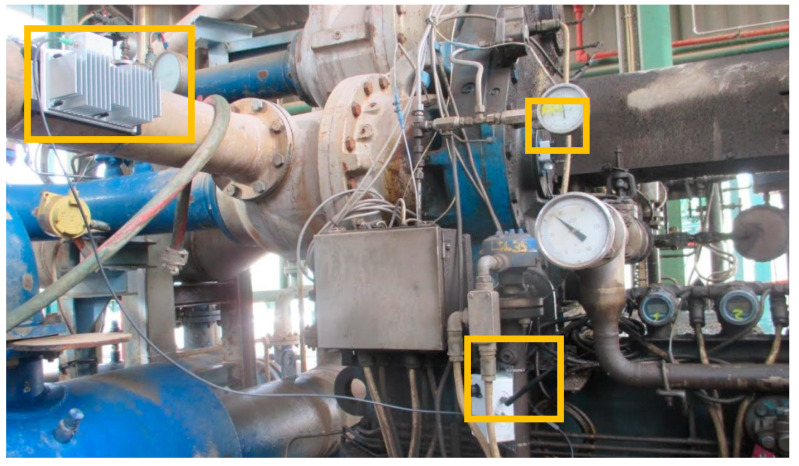
The complete installation of Repsol Puertallano’s oil refinery air compressor.

**Figure 17 sensors-25-00703-f017:**
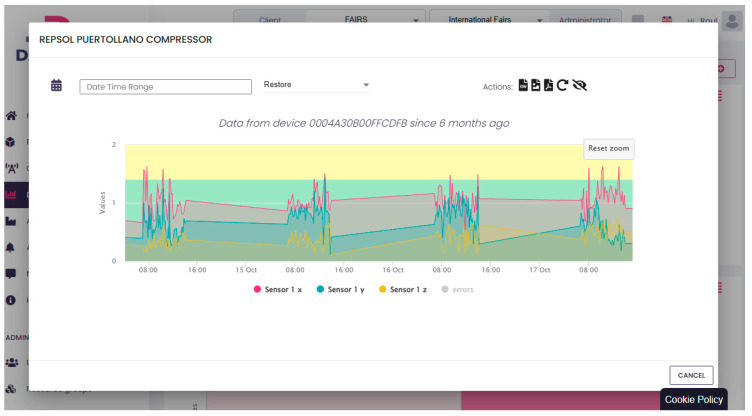
Results of four-day test in the combustion fan.

**Table 1 sensors-25-00703-t001:** Selection of industrial environments in which waste heat is present.

Type of Industry	Process Used	Temperature Range (°C)
**Iron and steel production;** **ferrous metals processing**	Basic oxygen Steel making	200
	Re-heating and heat treatment furnaces radiation	240
**Cement manufacturing**	Steam and gas exhausts	130–220
	Co-generation/combined heat and power	100
**Chemical and petrochemical** **large-volume inorganic chemicals–solids industry**	Sulphur burning process	145
**Chemical and petrochemical** **Large-volume inorganic chemicals—ammonia, acids, and fertilizers**	Conventional steam reforming–desulphurization process	350–400
	Conventional steam reforming—primary and secondary reforming	Primary: 400–600 Secondary: 400–600 Exhaust gas: 1000
**Chemical and petrochemical** **surface treatment using organic solvents**	Drying and curing	300–700
	Manufacturing of abrasives	35–110 in the drier 700 for the exhaust air
	Coil coating	150–220
**Food and tobacco;** **Food, drink, and milk industry**	Heat recovery from cooling systems	50–60
	Winery exhausts	200–240
	Alcohol distillation exhausts	130–220
**Wood;** **wood-based panels production**	Drying of wood particles	60–220
	Pressing	100–300
**Paper, pulp, print, and board production**	Papermaking and related processes	150–300 (combustion exhausts) >350 (coated wood-free printing tissue process with conv. Yankee dryer)
**Textile and leather industry**	Tanning and hides drying	60–90
	Drying	130
**Non-specific industry**	Drying and degassing	100–300
	Drying	100
	Drying of wood particles	200–370 for single/triple-pass dryers 500 for rotary dryers

**Table 2 sensors-25-00703-t002:** LPWAN comparison.

LPWAN Techno	SCADA Integration	ATEX/IECEX Compliant	Spectrum	Freq.	Max Data Rate	Range (km)
**SigFox**	No	Yes	Unlic.	Regional sub-GHz bands 868/902 MHz	100 bps	3~17
**LoRaWAN**	Yes	Yes	Unlic.	Regional sub-GHz bands 433/780/868/915 MHz	50 kbps	2~14
**LTE-M.**	No	No	Lic.	LTE in-bands only 1.08/1.4 GHz	1 Mbps	~11
**NB-IoT**	No	No	Lic.	LTE in-band 900 MHz	256 kbps	~22
**Zigbee**	Limited	No	Unlic.	2.4 GHz	250 Kbps	0.01–0.1
**Wifi**	Yes	Yes	Unlic.	2.4/5/6 GHz	9.6 Gbps	0.05–0.1
**Bluetooth**	Limited	No	Unlic.	2.4 Ghz	3 Mbps	0.01–0.1 depending on class

**Table 3 sensors-25-00703-t003:** HEAT-RU power generation.

Time (h)	Tpipe (°C)	Thot (°C)	Tcold (°C)	ΔT (°C)	Pgen. (W)
4548	200	172	123	49	0.91
4567	189	150	107	43	0.81
4589	190	163	114	49	0.92
4594	157	132	92	40	0.78
4597	167	140	96	44	0.83
4601	167	140	96	44	0.94
4623	197	163	106	57	1.05

**Table 4 sensors-25-00703-t004:** Power comparison.

**Protocols**	**NB-IoT**	**LTE-CATM1**	**LoRaWAN**
UDP	1.17 mWh	2.15 mWh	--
TCP	1.73 mWh	2.71 mWh	--
MQTT	1.71 mWh	2.45 mWh	76, 6u Wh
MQTT-TLS	2.82 mWh	2.64 mWh	--
**Protocols**	**NB-IoT**	**LTE-CATM1**	**LoRaWAN**
UDP	1.17 mWh	2.15 mWh	--
TCP	1.73 mWh	2.71 mWh	--
MQTT	1.71 mWh	2.45 mWh	76, 6u Wh
MQTT-TLS	2.82 mWh	2.64 mWh	--

**Table 5 sensors-25-00703-t005:** Comparison of commercial vibration sensors.

Product/Model	Wireless Technology	Energy Source	Bandwidth	No. Axis	Information Sent	Data Transmit
**AEInnova Indueye** **(this paper)**	NB-IOT	Waste heat	1 Hz to 2 KHz	3	Velocity 3 axis Spectrum	Every 60 s up to 1 h
**Everactive/Fluke 3562** [26]	Proprietary Wireless protocol. Up to 1000 nodes and 250 m.	Waste heat or solar	6 Hz to 1 KHz	3	Velocity 3 axis Spectrum	From 15 s to 15 min
**Emerson AMS** [28]	WirelessHART. Up to 100 nodes and 50 m.	Battery; expected 3–5 years in lab conditions	X, Y 1 KHz, Z up to 20 KHz	3	Velocity 3 axis Spectrum Battery voltage	1 h velocity, 1 time per day spectrum
**SKF Vibration Sensor** [29]	WirelessHART. Up to 100 nodes and 50 m.	Battery; expected 2–3 years in lab conditions	10 Hz to 1 KHz	1	Vibration: Velocity 3 axis Acceleration Battery voltage Temperature: −40 to 85 °C Precision +/−2 °C	Temperature every 5 min; vibration every 1 h
**Yokogawa Sushi** [30]	**LoRaWAN (up to 2 km)**	**Battery; expected 2 years**	**10 Hz to 1 KHz**	**3**	**Vibration:** **Velocity 3 axis** **Acceleration** **Battery voltage** **Temperature:** **−20 °C to 85 °C**	**1 datum per min up to every 3 days**

## Data Availability

All data are available from public funding agencies: Agencia Estatal de investigación (https://www.aei.gob.es/ (accessed on 21 October 2024)), European Commission EIC Accelerator (https://eic.ec.europa.eu/eic-funding-opportunities/eic-accelerator_en (accessed on 21 October 2024)), and AEInnova ( https://aeinnova.com/proyectos/ (accessed on 21 October 2024)).

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
