# Peer review of "Transforming Industrial Maintenance with Thermoelectric Energy Harvesting and NB-IoT: A Case Study in Oil Refinery Applications"

_sensors, 2025, doi:10.3390/s25030703_

Round 1
Reviewer 1 Report
Comments and Suggestions for Authors
In this paper, to address the problem of shortened battery life of wireless devices in high-temperature environments and high maintenance costs due to incompatibility with explosive environments in energy-intensive and explosive industries, the authors design a thermoelectric power generation system that utilizes waste heat to power ATEX-certified IIoT devices, which effectively reduces the significant energy loss due to waste heat in industries. Also, the authors analyze the performance of IIoT vibration sensors based on NB-IoT wireless protocol and evaluate the impact of MQTT+TLS and UDP on the system power consumption. Finally, the system performance is summarized based on the numerical analysis discussion. However, there are still some issues for improvement:
(1)In the experimental part, the comparative analysis of NB-IoT protocol with LoRa, wifi, ZigBee and other protocols is missing, and more graphs should be added to illustrate the effectiveness of NB-IoT protocol;
(2)In the experimental part, it is suggested to supplement the performance comparison between this system and other solutions in the market to enhance the persuasive power of the paper;
(3)Tables 3 and 4 in the text are incorrectly formatted;
(4)Adjust the thesis picture layout format and control the picture size. The figure note on page 8 of the text is incorrect, the figure and the figure note in Figure 7 need to be on one page, and Figure 11, Figure 13, etc. need to be centered;
(5)Some of the references are incorrectly formatted and need further checking and proofreading;
Author Response
COMMENTS 1:
In the experimental part, the comparative analysis of NB-IoT protocol with LoRa, wifi, ZigBee and other protocols is missing, and more graphs should be added to illustrate the effectiveness of NB-IoT protocol;
RESPONSE 1:
Dear reviewer, thanks for your comment. In this paper it has been only implemented the hardware using NB-IoT protocol, designing a new complete board. The hardware has no “shields” to interchange from NB-IoT to other low range and long ranges chipsets. In this case my proposal is to add in the table 2 (in section 4) some more protocols to compare with NB-IOT), and a bit more analysis rewriting lines from 423 to 426, and the following paragraph of Table 2.
Moreover, like in a previous work presented in MDPI Information the whole system works with LoRAWAN, I have cited this work [27], and introduced its energy profile (new figure 13), adding a new column in Table 4 with the loraWan energy profile. This comparison is aligned with comments 2.
They are market in RED. I hope that now is more understandable.
COMMENTS 2:
In the experimental part, it is suggested to supplement the performance comparison between this system and other solutions in the market to enhance the persuasive power of the paper;
RESPONSE 2:
Thanks for pointing out. Accordingly with comments 1, we have added the results of a previous work using loraWAN.
Additionally we have added a new subsection called 5.5 related work with a complete comparison with commercial competitors. I think this contribution creates a better understanding of the work. Thanks.
COMMENTS 3:
Tables 3 and 4 in the text are incorrectly formatted;
RESPONSE 3:
You are right. According with (https://www.mdpi.com/journal/sensors/instructions) we have modied the format. Thanks.
COMMENTS 4:
Adjust the thesis picture layout format and control the picture size. The figure note on page 8 of the text is incorrect, the figure and the figure note in Figure 7 need to be on one page, and Figure 11, Figure 13, etc. need to be centered;
RESPONSE 4:
Thanks for your comments. We have modifed the notes of these figures and centered new ones. Also we saw that the numeration of several figures were wrong, so we have renumbered them. About other “size” or formats of figure, the editor of the paper will perform the “final changes” to fit in the format. Thanks!
COMMENTS 5.
Some of the references are incorrectly formatted and need further checking and proofreading
RESPONSE 5:
Thank you for your valuable feedback. I have modified reference’s format (all in red). Also the editor will make a last review of them.
Once again, I sincerely appreciate your time and effort in reviewing this paper. I hope this revised version meets your expectations.
Reviewer 2 Report
Comments and Suggestions for Authors
I appreciated the conducted work, especially the industry-oriented approach, and the manner in which the paper is written.
I have some comments that in essence should set the boundaries for the presented work:
- There is no figure 1 as it is expressed at line 46
- It is arguable if batteries are the main energy source for IIoT devices. I consider that the statement has to be nuanced and highlighted only to wireless networks without local power wiring, because most of the IIoT devices are relying on direct connection to the local energy network (even if wireless data transmission is used) and do not contain batteries at all. Also, in some sectors the local wireless communication has limited or no coverage yet. Although, the idea and the purpose of the study is clear and sound.
- Also, another aspect has to be clearly specified, related to the usage of NB-IoT. The ISA95 hierarchy is currently used in critical domains, and due to the current security concerns data is not sent directly to the cloud, but connected directly to PLCs or SCADA systems. Therefore, if only NB-IoT is the target of the device, it has to be clearly stated that the device targets only the specific architecture from fig. 8. Also, the architecture from the interfacing/SCADA perspective has to be clearly specified from the applicability point of view. If the device targets also other 5-7 OSI level Industry 4.0 protocols (besides the LPWAN specific ones) is has to be mentioned at least as development perspective. The targeted protocols may also influence the energy consumption as the authors are analyzing correctly for below OSI 5 level protocols.
Author Response
COMMENTS 1:
There is no figure 1 as it is expressed at line 46
RESPONSE 1:
Dear reviewer. First of all thanks for accepting the review of this paper.
You are right, finally the figure will not be include due to authors’ copyright. In the reference [6] appear the figure and all information. I have modified the paragraph.
COMMENTS 2:
- It is arguable if batteries are the main energy source for IIoT devices. I consider that the statement has to be nuanced and highlighted only to wireless networks without local power wiring, because most of the IIoT devices are relying on direct connection to the local energy network (even if wireless data transmission is used) and do not contain batteries at all. Also, in some sectors the local wireless communication has limited or no coverage yet. Although, the idea and the purpose of the study is clear and sound.
RESPONSE 2:
Thank you for your insightful comment. You are indeed correct that many IIoT devices rely on direct connection to the local energy network and may not require batteries, especially when wireless data transmission is used in areas with sufficient infrastructure. However, as you rightly noted, local wireless communication can still face significant coverage limitations in certain sectors.
In this context, long-range wireless protocols such as LoRaWAN and NB-IoT—the latter being the focus of this study—are rapidly emerging as key technologies to address this gap. These protocols enable reliable data transmission in remote areas of large industrial plants where wired connections are impractical or unavailable, while reducing dependency on local power infrastructure.
NB-IoT, in particular, leverages existing cellular networks (3G, 4G) to provide extended coverage and penetration, making it a promising solution for remote monitoring applications where both power availability and network access are limited.
We will adjust the manuscript to clarify this distinction and ensure a more precise description of battery usage within wireless IIoT systems. Thank you again for your valuable feedback, which helps us strengthen the accuracy and clarity of our work.
We have replaced the paragraph in lines 88 to 91 to a new text that contextualised your comment. Now the text is located from 88 to 95.
Old text:
Energy-intensive industries (EIIs) face a significant challenge in improving their operational efficiency, maintenance processes, energy consumption, and overall sustainability. This section will study a potential technology to power a new generation of heat-powered wireless devices.
New text:
Energy-intensive industries (EIIs) face a significant challenge in improving their operational efficiency, maintenance processes, energy consumption, and overall sustainability. While many IIoT devices rely on direct power connections, the increasing adoption of long-range wireless protocols such as LoRaWAN and NB-IoT—the latter being the focus of this study—offers a transformative solution for remote monitoring in areas where local power wiring is unavailable or impractical. These protocols not only enable reliable data transmission but also open the door to battery-free, heat-powered devices, contributing to enhanced energy efficiency and sustainability.
COMMENTS 3:
- Also, another aspect has to be clearly specified, related to the usage of NB-IoT. The ISA95 hierarchy is currently used in critical domains, and due to the current security concerns data is not sent directly to the cloud, but connected directly to PLCs or SCADA systems. Therefore, if only NB-IoT is the target of the device, it has to be clearly stated that the device targets only the specific architecture from fig. 8. Also, the architecture from the interfacing/SCADA perspective has to be clearly specified from the applicability point of view. If the device targets also other 5-7 OSI level Industry 4.0 protocols (besides the LPWAN specific ones) is has to be mentioned at least as development perspective. The targeted protocols may also influence the energy consumption as the authors are analyzing correctly for below OSI 5 level protocols.
RESPONSE 3:
Thank you for your valuable feedback regarding the use of NB-IoT in the context of the ISA95 hierarchy and SCADA systems. You are absolutely right that, in critical domains, data security concerns often limit direct cloud connectivity, with most IIoT data being processed through PLCs or SCADA platforms using well-established protocols such as OPC UA, Modbus TCP, or EtherCAT.
However, one of the motivations behind the use of NB-IoT in this study is its ability to facilitate direct cloud connectivity for advanced data analytics applications, including Machine Learning (ML) and Deep Learning (DL), which are often challenging to implement in traditional SCADA architectures due to data silos and limited cloud integration.
While we acknowledge the importance of on-premises monitoring through SCADA, our proposed solution focuses on scenarios where remote condition monitoring and cloud-based analytics are essential, even if this could present integration challenges with existing SCADA systems. We will revise the manuscript to clarify that the device architecture specifically targets the cloud-first approach depicted in old Figure 8 (now figure 11), while recognizing the relevance of other OSI layers 5-7 protocols as part of future development considerations.
Particularlary we have added the following paragraph between line 96 and 99:
This work focuses on NB-IoT, a protocol that enables direct cloud connectivity, facilitating advanced data analytics, including Machine Learning (ML) and Deep Learning (DL), often limited in traditional SCADA systems.
We have added a note below the figure 11 with this text:
Note: The architecture shown in Figure 11 targets a cloud-first approach using the NB-IoT protocol, which may face integration challenges with existing SCADA systems based on ISA95. The current design focuses on remote data acquisition for cloud-based analytics but can be adapted for local SCADA integration using additional industry-standard protocols (e.g., OPC UA, Modbus TCP, EtherCAT) in future iterations.
And in the middle of the conclusions:
While the primary focus of this study is on remote data transmission for cloud-based analytics using NB-IoT, we acknowledge that most critical industrial environments rely on direct SCADA integration due to security concerns under the ISA95 hierarchy. Future developments could explore multi-protocol compatibility to facilitate both cloud and on-premises data management.
Thank you again for your constructive feedback, which will help us better position the scope and applicability of the proposed solution.